# Regularization in ResNet with Stochastic Depth

**Soufiane Hayou**[*]
Department of Statistics
University of Oxford
United Kingdom

**Fadhel Ayed**[*]
Huawei Technologies
France

## Abstract

Regularization plays a major role in modern deep learning. From classic techniques such as $L_1, L_2$ penalties to other noise-based methods such as Dropout, regularization often yields better generalization properties by avoiding overfitting. Recently, Stochastic Depth ($\mathcal{SD}$) has emerged as an alternative regularization technique for residual neural networks (ResNets) and has proven to boost the performance of ResNet on many tasks [Huang et al., 2016]. Despite the recent success of $\mathcal{SD}$, little is known about this technique from a theoretical perspective. This paper provides a hybrid analysis combining perturbation analysis and signal propagation to shed light on different regularization effects of $\mathcal{SD}$. Our analysis allows us to derive principled guidelines for choosing the survival rates used for training with $\mathcal{SD}$.

## 1 Introduction

Stochastic Depth ($\mathcal{SD}$) is a well-established regularization method that was first introduced by Huang et al. [2016]. It is similar in principle to Dropout [Hinton et al., 2012, Srivastava et al., 2014] and DropConnect [Wan et al., 2013]. It belongs to the family of noise-based regularization techniques, which includes other methods such as noise injection in data [Webb, 1994, Bishop, 1995] and noise injection throughout the network [Camuto et al., 2020]. While Dropout, resp. DropConnect consists of removing some neurons, resp. weights, at each iteration, $\mathcal{SD}$ randomly drops *full layers*, and only updates the weights of the resulting subnetwork at each training iteration. As a result of this mechanism, $\mathcal{SD}$ can be exclusively used with residual neural networks (ResNets).

There exists a stream of papers in the literature on the regularization effect of Dropout for linear models [Wager et al., 2013, Mianjy and Arora, 2019, Helmbold and Long, 2015, Cavazza et al., 2017]. Recent work by Wei et al. [2020] extended this analysis to deep neural networks using second-order perturbation analysis. It disentangled the explicit regularization of Dropout on the loss function and the implicit regularization on the gradient. Similarly, Camuto et al. [2020] studied the explicit regularization effect induced by adding Gaussian Noise to the activations and empirically illustrated the benefits of this regularization scheme. However, to the best of our knowledge, no analytical study of $\mathcal{SD}$ exists in the literature. This paper aims to fill this gap by studying the regularization effect of $\mathcal{SD}$ from an analytical point of view; this allows us to derive principled guidelines on the choice of the survival probabilities for network layers. Concretely, our contributions are four-fold:

- We show that $\mathcal{SD}$ acts as an explicit regularizer on the loss function by penalizing a notion of *information discrepancy* between keeping and removing certain layers.

- We prove that the *uniform mode*, defined as the choice of constant survival probabilities, is related to maximum regularization using $\mathcal{SD}$.

- We study the large depth behaviour of $\mathcal{SD}$ and show that in this limit, $\mathcal{SD}$ mimics *Gaussian Noise Injection* by implicitly adding data-adaptive Gaussian noise to the pre-activations.

---

[*]Equal contribution. Correspondence to: <soufiane.hayou@yahoo.fr; fadhel.ayed@huawei.com>

35th Conference on Neural Information Processing Systems (NeurIPS 2021).

- By defining the training budget $\bar{L}$ as the *desired* average depth, we show the existence of two different regimes: *small budget* and *large budget* regimes. We introduce a new algorithm called *SenseMode* to compute the survival rates under a fixed training budget and provide a series of experiments that validates our *Budget hypothesis* introduced in Section 5.

## 2   Stochastic Depth Neural Networks

Stochastic depth neural networks were first introduced by Huang et al. [2016]. They are standard residual neural networks with random depth. In practice, each block in the residual network is multiplied by a random Bernoulli variable $\delta_l$ ($l$ is the block's index) that is equal to $1$ with some survival probability $p_l$ and $0$ otherwise. The mask is re-sampled after each training iteration, making the gradient act solely on the subnetwork composed of blocks with $\delta_l = 1$.

We consider a slightly different version where we apply the binary mask to the pre-activations instead of the activations. We define a depth $L$ stochastic depth ResNet by

$$
\begin{aligned}
y_0(x; \boldsymbol{\delta}) &= \Psi_0(x, W_0), \\
y_l(x; \boldsymbol{\delta}) &= y_{l-1}(x; \boldsymbol{\delta}) + \delta_l \Psi_l(y_{l-1}(x; \boldsymbol{\delta}), W_l), \quad 1 \le l \le L, \\
y_{out}(x; \boldsymbol{\delta}) &= \Psi_{out}(y_L(x; \boldsymbol{\delta}), W_{out}),
\end{aligned}
\tag{1}
$$

where $W_l$ are the weights in the $l^{th}$ layer, $\Psi$ is a mapping that defines the nature of the layer, $y_l$ are the pre-activations, and $\boldsymbol{\delta} = (\delta_l)_{1 \le l \le L}$ is a vector of Bernoulli variables with survival parameters $\boldsymbol{p} = (p_l)_{1 \le l \le L}$. $\boldsymbol{\delta}$ is re-sampled at each iteration. For the sake of simplification, we consider constant width ResNet and we further denote by $N$ the width, i.e. for all $l \in [L-1]$, $y_l \in \mathbb{R}^N$. The output function of the network is given by $s(y_{out})$ where $s$ is some convenient mapping for the learning task, e.g. the Softmax mapping for classification tasks. We denote by $o$ the dimension of the network output, i.e. $s(y_{out}) \in \mathbb{R}^o$ which is also the dimension of $y_{out}$. For our theoretical analysis, we consider a Vanilla model with residual blocks composed of a Fully Connected linear layer

$$
\Psi_l(x, W) = W\phi(x),
$$

where $\phi(x)$ is the activation function. The weights are initialized with He init [He et al., 2015], e.g. for ReLU, $W_{ij}^l \sim \mathcal{N}(0, 2/N)$.

There are no principled guidelines on choosing the survival probabilities. However, the original paper by Huang et al. [2016] proposes two alternatives that appear to make empirical consensus: the *uniform* and *linear modes*, described by

$$
\textbf{Uniform: } p_l = p_L, \quad \textbf{Linear: } p_l = 1 - \tfrac{l}{L}(1 - p_L),
$$

where we conveniently parameterize both alternatives using $p_L$.

**Training budget $\bar{L}$.** We define the training budget $\bar{L}$ to be the desired average depth of the subnetworks with $\mathcal{SD}$. The user typically fixes this budget, e.g., a small budget can be necessary when training is conducted on small capacity devices.

**Depth of the subnetwork.** Given the mode $\boldsymbol{p} = (p_l)_{1 \le l \le L}$, after each iteration, the subnetwork has a depth $L_{\boldsymbol{\delta}} = \sum_{l=1}^{L} \delta_l$ with an average $L_{\boldsymbol{p}} := \mathbb{E}_{\boldsymbol{\delta}}[L_{\boldsymbol{\delta}}] = \sum_{l=1}^{L} p_l$. Given a budget $\bar{L}$, there is a infinite number of modes $\boldsymbol{p}$ such that $L_{\boldsymbol{p}} = \bar{L}$. In the next lemma, we provide probabilistic bounds on $L_{\boldsymbol{\delta}}$ using standard concentration inequalities. We also show that with a fixed budget $\bar{L}$, the uniform mode is linked to maximal variability.

**Lemma 1** (Concentration of $L_{\boldsymbol{\delta}}$). *For any $\beta \in (0, 1)$, we have that with probability at least $1 - \beta$,*

$$
|L_{\boldsymbol{\delta}} - L_{\boldsymbol{p}}| \le v_{\boldsymbol{p}} \, u^{-1}\left( \frac{\log(2/\beta)}{v_{\boldsymbol{p}}} \right),
\tag{2}
$$

*where $L_{\boldsymbol{p}} = \mathbb{E}[L_{\boldsymbol{\delta}}] = \sum_{l=1}^{L} p_l$, $v_{\boldsymbol{p}} = \mathrm{Var}[L_{\boldsymbol{\delta}}] = \sum_{l=1}^{L} p_l(1 - p_l)$, and $u(t) = (1+t)\log(1+t) - t$.*

*Moreover, for a given average depth $L_{\boldsymbol{p}} = \bar{L}$, the upperbound in Eq. (2) is maximal for the uniform choice of survival probabilities $\boldsymbol{p} = \left( \frac{\bar{L}}{L}, ..., \frac{\bar{L}}{L} \right)$.*

Lemma 1 shows that with high probability, the depth of the subnetwork that we obtain with $\mathcal{SD}$ is within an $\ell_1$ error of $v_{\boldsymbol{p}}\,u^{-1}\left(\frac{\log(2/\beta)}{v_{\boldsymbol{p}}}\right)$ from the average depth $L_{\boldsymbol{p}}$. Given a fixed budget $\bar{L}$, this segment is maximized for the uniform mode $\boldsymbol{p} = (\bar{L}/L, \dots, \bar{L}/L)$. Fig. 1a highlights this result. This was expected since the variance of the depth $L_{\boldsymbol{\delta}}$ is also maximized by the uniform mode. The uniform mode corresponds to maximum entropy of the random depth, which would intuitively results in maximum regularization. We depict this behaviour in more details in Section 4.

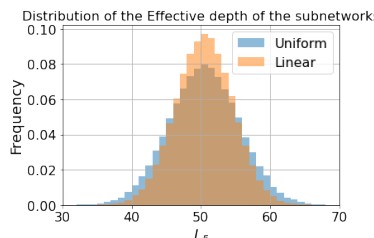

(a) Distributions of $L_{\boldsymbol{\delta}}$ for a Resnet100 with average survival rate $\bar{L}/L = 0.5$ for the uniform and linear modes.

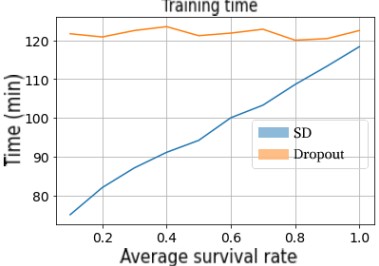

(b) Training time of Dropout and $\mathcal{SD}$ on CIFAR10 with ResNet56 for 100 epochs.

Figure 1

**$\mathcal{SD}$ vs Dropout.** From a computational point of view, $\mathcal{SD}$ has the advantage of reducing the effective depth during training. Depending on the chosen budget, the subnetworks might be significantly shallower than the entire network (Lemma 1). This depth compression can be effectively leveraged for computational training time gain (Fig. 1b). It is not the case with Dropout. Indeed, assuming that the choice of dropout probabilities is such that we keep the same number of parameters on average compared to $\mathcal{SD}$, we still have to multiply matrices $L$ times during the forward/backward propagation. It is not straightforward to leverage the sparsity obtained by Dropout for computational gain. In practice, Dropout requires an additional step of sampling the mask for every neuron, resulting in longer training times than without Dropout (Fig. 1b). However, there is a trade-off between how small the budget is and the performance of the trained model with $\mathcal{SD}$ (Section 6).

## 3 Effect of Stochastic Depth at initialization

Empirical evidence strongly suggests that Stochastic Depth allows training deeper models [Huang et al., 2016]. Intuitively, at each iteration, $\mathcal{SD}$ updates only the parameters of a subnetwork with average depth $L_{\boldsymbol{p}} = \sum_l p_l < L$, which could potentially alleviate any exploding/vanishing gradient issue. This phenomenon is often faced when training ultra deep neural networks. To formalize this intuition, we consider the model's asymptotic properties at initialization in the infinite-width limit $N \to +\infty$. This regime has been the focus of several theoretical studies [Neal, 1995, Poole et al., 2016, Schoenholz et al., 2017, Yang, 2020, Xiao et al., 2018, Hayou et al., 2019, 2020, 2021a] since it allows to derive analytically the distributions of different quantities of untrained neural networks. Specifically, randomly initialized ResNets, as well as other commonly-used architectures such as Fully connected Feedforward networks, convolutional networks and LSTMs, are equivalent to Gaussian Processes in the infinite-width limit. An important ingredient in this theory is the *Gradient Independence* assumption. Let us formally state this assumption first.

**Assumption 1** (Gradient Independence). *In the infinite width limit, we assume that the weights $\boldsymbol{W}$ used for back-propagation are an iid version of the weights $\boldsymbol{W}$ used for forward propagation.*

Assumption 1 is ubiquitous in the literature on the signal propagation in deep neural networks. It has been used to derive theoretical results on signal propagation in randomly initialized deep neural network [Schoenholz et al., 2017, Poole et al., 2016, Yang and Schoenholz, 2017, Hayou et al., 2021b,a] and is also a key tool in the derivation of the Neural Tangent Kernel [Jacot et al., 2018, Arora et al., 2019, Hayou et al., 2020]. Recently, it has been shown by Yang [2020] that Assumption 1 yields the exact computation for the gradient covariance in the infinite width limit. See Appendix A0.3 for a detailed discussion about this assumption. Throughout the paper, we provide numerical results that substantiate the theoretical results that we derive using this assumption. We show that Assumption 1 yields an excellent match between theoretical results and numerical experiments.

Leveraging this assumption, Yang and Schoenholz [2017], Hayou et al. [2021a] proved that ResNet suffers from exploding gradient at initialization. We show in the next proposition that $\mathcal{SD}$ helps mitigate the exploding gradient behaviour at initialization in infinite width ResNets.

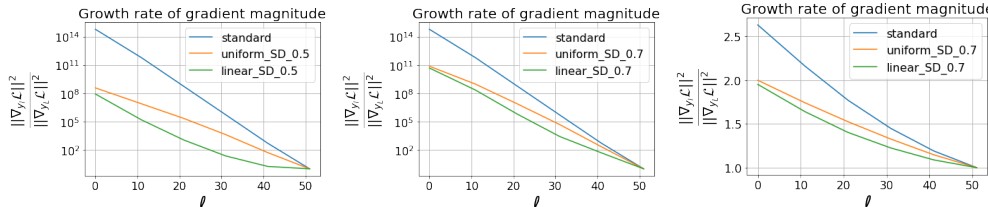

(a) $\bar{L}/L = 0.5$, standard ResNet.    (b) $\bar{L}/L = 0.7$, standard ResNet.    (c) $\bar{L}/L = 0.7$, stable ResNet.

Figure 2: Empirical illustration of Proposition 1 ((a) and (b)) and Stable Resnet (c). Comparison of the growth rate of the gradient magnitude $\tilde{q}_l(x, z)$ at initialization for Vanilla ResNet50 with width 512. The y-axis of figures (a) and (b) are in log scale. The y-axis of figure (c) is in linear scale. The expectation is computed using 500 Monte-Carlo (MC) samples.

**Proposition 1.** *Let $\phi = ReLU$ and $\mathcal{L}(x, z) = \ell(y_{out}(x; \boldsymbol{\delta}), z)$ for $(x, z) \in \mathbb{R}^d \times \mathbb{R}^o$, where $\ell(z, z')$ is some differentiable loss function. Let $\tilde{q}_l(x, z) = \mathbb{E}_{W,\boldsymbol{\delta}} \frac{\|\nabla_{y_l}\mathcal{L}\|^2}{\|\nabla_{y_L}\mathcal{L}\|^2}$, where the numerator and denominator are respectively the norms of the gradients with respect to the inputs of the $l^{th}$ and $L^{th}$ layers. Then, in the infinite width limit, under Assumption 1, for all $l \in [L]$ and $(x, z) \in \mathbb{R}^d \times \mathbb{R}^o$, we have*

- *With Stochastic Depth, $\tilde{q}_l(x, z) = \prod_{k=l+1}^{L}(1 + p_k)$,*

- *Without Stochastic Depth (i.e. $\boldsymbol{\delta} = \mathbf{1}$), $\tilde{q}_l(x, z) = 2^{L-l}$.*

Proposition 1 indicates that with or without $\mathcal{SD}$, the gradient explodes exponentially at initialization as it backpropagates through the network. However, with $\mathcal{SD}$, the exponential growth is characterized by the mode $\boldsymbol{p}$. Intuitively, if we choose $p_l \ll 1$ for some layer $l$, then the contribution of this layer in the exponential growth is negligible since $1 + p_l \approx 1$. From a practical point of view, the choice of $p_l \ll 1$ means that the $l^{th}$ layer is hardly present in any subnetwork during training, thus making its contribution to the gradient negligible on average (w.r.t $\boldsymbol{\delta}$). For a ResNet with $L = 50$ and uniform mode $\boldsymbol{p} = (1/2, \ldots, 1/2)$, $\mathcal{SD}$ reduces the gradient exploding by six orders of magnitude. Fig. 2a and Fig. 2b

Table 1: Gradient magnitude growth rate with Vanilla Resnet50 with width 512 and training budget $\bar{L}/L = 0.7$. Empirical vs. Theoretical value (between parenthesis) of $\tilde{q}_l(x, z)$ at initialization, for standard (no $\mathcal{SD}$), uniform and linear modes. The expectation is performed using 500 MC samples.

| $\ell$ | Standard | Uniform | Linear |
|---|---|---|---|
| 0 | 2.001 (2) | 1.705 (1.7) | 1.694 (1.691) |
| 10 | 2.001 (2) | 1.708 (1.7) | 1.633 (1.629) |
| 20 | 2.001 (2) | 1.707 (1.7) | 1.569 (1.573) |
| 30 | 2.001 (2) | 1.716 (1.7) | 1.555 (1.516) |
| 40 | 1.999 (2) | 1.739 (1.7) | 1.530 (1.459) |

illustrates the exponential growth of the gradient for the uniform and linear modes, as compared to the growth of the gradient without $\mathcal{SD}$. We compare the empirical/theoretical growth rates of the magnitude of the gradient in Table 1; the results show a good match between our theoretical result (under Assumption 1) and the empirical ones. Further analysis can be found in Section A4.

**Stable ResNet.** Hayou et al. [2021a] have shown that introducing the scaling factor $1/\sqrt{L}$ in front of the residual blocks is sufficient to avoid the exploding gradient at initialization, as illustrated in Figure 2c. The hidden layers in Stable ResNet (with $\mathcal{SD}$) are given by,

$$y_l(x; \boldsymbol{\delta}) = y_{l-1}(x; \boldsymbol{\delta}) + \frac{\delta_l}{\sqrt{L}}\Psi_l(y_{l-1}(x; \boldsymbol{\delta}), W_l), \quad 1 \leq l \leq L. \tag{3}$$

The intuition behind the choice of the scaling factor $1/\sqrt{L}$ comes for the variance of $y_l$. At initialization, with standard ResNet (Eq. (1)), we have $\text{Var}[y_l] = \text{Var}[y_{l-1}] + \Theta(1)$, which implies that $\text{Var}[y_l] = \Theta(l)$. With Stable ResNet (Eq. (3)), this becomes $\text{Var}[y_l] = \text{Var}[y_{l-1}] + \Theta(1/L)$, resulting in $\text{Var}[y_l] = \Theta(1)$ (See Hayou et al. [2021a] for more details). In the rest of the paper, we restrict our analysis to Stable ResNet; this will help isolate the regularization effect of $\mathcal{SD}$ in the limit of large depth without any variance/gradient exploding issue.

Nevertheless, the natural connection between $\mathcal{SD}$ and Dropout, coupled with the line of work on the regularization effect induced by the latter [Wager et al., 2013, Mianjy and Arora, 2019, Helmbold and Long, 2015, Cavazza et al., 2017, Wei et al., 2020], would indicate that the benefits of $\mathcal{SD}$ are not limited to controlling the magnitude of the gradient. Using a second order Taylor expansion, Wei et al. [2020] have shown that Dropout induces an explicit regularization on the loss function. Intuitively, one should expect a similar effect with $\mathcal{SD}$. In the next section, we elucidate the *explicit* regularization effect of $\mathcal{SD}$ on the loss function, and we shed light on another regularization effect of $\mathcal{SD}$ that occurs in the large depth limit.

# 4 Regularization effect of Stochastic Depth

## 4.1 Explicit regularization on the loss function

Consider a dataset $\mathcal{D} = \mathcal{X} \times \mathcal{T}$ consisting of $n$ (input, target) pairs $\{(x_i, t_i)\}_{1 \leq i \leq n}$ with $(x_i, t_i) \in \mathbb{R}^d \times \mathbb{R}^o$. Let $\ell : \mathbb{R}^d \times \mathbb{R}^o \rightarrow \mathbb{R}$ be a smooth loss function, e.g. quadratic loss, cross-entropy loss etc. Define the model loss for a single sample $(x, t) \in \mathcal{D}$ by

$$\mathcal{L}(\boldsymbol{W}, x; \boldsymbol{\delta}) = \ell(y_{out}(x; \boldsymbol{\delta}), t), \quad \mathcal{L}(\boldsymbol{W}, x) = \mathbb{E}_\delta \left[\ell(y_{out}(x; \boldsymbol{\delta}), t)\right],$$

where $\boldsymbol{W} = (W_l)_{0 \leq l \leq L}$. The empirical loss given by $\mathcal{L}(\boldsymbol{W}) = \frac{1}{n} \sum_{i=1}^n \mathbb{E}_\delta \left[\ell(y_{out}(x_i; \boldsymbol{\delta}), t_i)\right]$.

To isolate the regularization effect of $\mathcal{SD}$ on the loss function, we use a second order approximation of the loss function around $\boldsymbol{\delta} = \mathbf{1}$, this allows us to marginalize out the mask $\boldsymbol{\delta}$. The full derivation is provided in Appendix A2. Let $z_l(x; \boldsymbol{\delta}) = \Psi_l(W_l, y_{l-1}(x; \boldsymbol{\delta}))$ be the activations. For some pair $(x, t) \in \mathcal{D}$, we obtain

$$\mathcal{L}(\boldsymbol{W}, x) \approx \bar{\mathcal{L}}(\boldsymbol{W}, x) + \frac{1}{2L} \sum_{l=1}^L p_l(1 - p_l)g_l(\boldsymbol{W}, x), \tag{4}$$

where $\bar{\mathcal{L}}(\boldsymbol{W}, x) \approx \ell(y_{out}(x; \boldsymbol{p}), t)$ (more precisely, $\bar{\mathcal{L}}(\boldsymbol{W}, x)$ is the second order Taylor approximation of $\ell(y_{out}(x; \boldsymbol{p}), t)$ around $\boldsymbol{p} = \mathbf{1}$[2]), and $g_l(\boldsymbol{W}, x) = z_l(x; \mathbf{1})^T \nabla_{y_l}^2 [\ell \circ G_l](y_l(x; \mathbf{1}))z_l(x; \mathbf{1})$ with $G_l$ is the function defined by $y_{out}(x; \mathbf{1}) = G_l(y_{l-1}(x; \mathbf{1}) + \frac{1}{\sqrt{L}} z_l(x; \mathbf{1}))$.

The first term $\bar{\mathcal{L}}(\boldsymbol{W}, x)$ in Eq. (4) is the loss function of the average network (i.e. replacing $\boldsymbol{\delta}$ with its mean $\boldsymbol{p}$). Thus, Eq. (4) shows that training with $\mathcal{SD}$ entails training the average network with an explicit regularization term that implicitly depends on the weights $\boldsymbol{W}$.

$\mathcal{SD}$ **enforces flatness.** The presence of the hessian in the penalization term provides a geometric interpretation of the regularization induced by $\mathcal{SD}$: it enforces a notion of flatness determined by the hessian of the loss function with respect to the hidden activations $z_l$. This flatness is naturally inherited by the weights, thus leading to flatter minima. Recent works by [Keskar et al., 2016, Jastrzebski et al., 2018, Yao et al., 2018] showed empirically that flat minima yield better generalization compared to minima with large second derivatives of the loss. Wei et al. [2020] have shown that a similar behaviour occurs in networks with Dropout.

Let $J_l(x) = \nabla_{y_l} G_l(y_l(x; \mathbf{1}))$ be the Jacobian of the output layer with respect to the hidden layer $y_l$ with $\boldsymbol{\delta} = \mathbf{1}$, and $H(x) = \nabla_z^2 \ell(z)_{|z=y_{out}(x; \mathbf{1})}$ the hessian of the loss function $\ell$. The hessian matrix inside the penalization terms $g_l(\boldsymbol{W}, x)$ can be decomposed as in [LeCun et al., 2012, Sagun et al., 2017]

$$\nabla_{y_l}^2 [\ell \circ G_l](y_l(x; \mathbf{1})) = J_l(x)^T H(x)J_l(x) + \Gamma_l(x),$$

where $\Gamma$ depends on the hessian of the network output. $\Gamma$ is generally non-PSD, and therefore cannot be seen as a regularizer. Moreover, it has been shown empirically that the first term generally dominates and drives the regularization effect [Sagun et al., 2017, Wei et al., 2020, Camuto et al., 2020]. Thus, we restrict our analysis to the regularization effect induced by the first term, and we consider the new version of $g_l$ defined by

$$g_l(\boldsymbol{W}, x) = \zeta_l(x, \boldsymbol{W})^T H(x) \zeta_l(x, \boldsymbol{W}) = \text{Tr}\left(H(x) \zeta_l(x, \boldsymbol{W})\zeta_l(x, \boldsymbol{W})^T\right), \tag{5}$$

where $\zeta_l(x, \boldsymbol{W}) = J_l(x)z_l(x; \mathbf{1})$. The quality of this approximation is discussed in Appendix A4.

**Information discrepancy.** The vector $\zeta_l$ represents a measure of the information discrepancy between keeping and removing the $l^{th}$ layer. Indeed, $\zeta_l$ measures the sensitivity of the model output to the $l^{th}$ layer,

$$y_{out}(x; \mathbf{1}) - y_{out}(x; \mathbf{1}_l) \approx \nabla_{\delta_l} y_{out}(x; \boldsymbol{\delta})_{|\boldsymbol{\delta}=\mathbf{1}} = \zeta_l(x, \boldsymbol{W}),$$

where $\mathbf{1}_l$ is the vector of $1's$ everywhere with $0$ in the $l^{th}$ coordinate.

With this in mind, the regularization term $g_l$ in Eq. (5) is most significant when the information discrepancy is well-aligned with the hessian of the loss function, i.e. $\mathcal{SD}$ penalizes mostly the layers with information discrepancy that violates the flatness, confirming our intuition above.

---

[2]Note that we could obtain Eq. (4) using the Taylor expansion around $\boldsymbol{\delta} = \boldsymbol{p}$. However, in this case, the Hessian will depend on $\boldsymbol{p}$, which complicates the analysis of the role of $\boldsymbol{p}$ in the regularization term.

**Quadratic loss.** With the quadratic loss $\ell(z, z') = \|z - z'\|_2^2$, the hessian $H_\ell(x) = 2I$ is isotropic, i.e. it does not favorite any direction over the others. Intuitively, we expect the penalization to be similar for all the layers. In this case, we have $g_l(\boldsymbol{W}, x) = 2 \|\zeta_l(x)\|_2^2$, and the loss is given by

$$\mathcal{L}(\boldsymbol{W}) \approx \bar{\mathcal{L}}(\boldsymbol{W}) + \frac{1}{2L} \sum_{l=1}^{L} p_l(1 - p_l)g_l(\boldsymbol{W}), \qquad (6)$$

where $g_l(\boldsymbol{W}) = \frac{2}{n} \sum_{i=1}^{n} \|\zeta_l(x_i, \boldsymbol{W})\|_2^2$ is the regularization term marginalized over inputs $\mathcal{X}$.

Eq. (6) shows that the mode $\boldsymbol{p}$ has a direct impact on the regularization term induced by $\mathcal{SD}$. The latter tends to penalize mostly the layers with survival probability $p_l$ close to $50\%$. The mode $\boldsymbol{p} = (1/2, \ldots, 1/2)$ is therefore a universal maximizer of the regularization term, given fixed weights $\boldsymbol{W}$. However, given a training budget $\bar{L}$, the mode $\boldsymbol{p}$ that maximizes the regularization term $\frac{1}{2L} \sum_{l=1}^{L} p_l(1 - p_l)g_l(W)$ depends on the values of $g_l(\boldsymbol{W})$. We show this in the next lemma.

**Lemma 2** (Max regularization). *Consider the empirical loss $\mathcal{L}$ given by Eq. (6) for some fixed weights $\boldsymbol{W}$ (e.g. $\boldsymbol{W}$ could be the weights at any training step of SGD). Then, given a training budget $\bar{L}$, the regularization is maximal for $p_l^* = \min\left(1, \max(0, \frac{1}{2} - Cg_l(\boldsymbol{W})^{-1})\right)$, where $C$ is a normalizing constant, that has the same sign as $L - 2\bar{L}$. The global maximum is obtained for $p_l = 1/2$.*

Lemma 2 shows that under fixed budget, the mode $\boldsymbol{p}^*$ that maximizes the regularization induced by $\mathcal{SD}$ is generally layer-dependent ($\neq$ uniform). However, we show that at initialization, on average (w.r.t $\boldsymbol{W}$), $\boldsymbol{p}^*$ is uniform.

**Theorem 1** ($p^*$ is uniform at initialization). *Assume $\phi = ReLU$ and $\boldsymbol{W}$ are initialized with $\mathcal{N}(0, \frac{2}{N})$. Then, in the infinite width limit, under Assumption 1, for all $l \in [1 : L]$, we have*

$$\mathbb{E}_{\boldsymbol{W}}[g_l(\boldsymbol{W})] = \mathbb{E}_{\boldsymbol{W}}[g_1(\boldsymbol{W})].$$

*As a result, given a budget $\bar{L}$, the average regularization term $\frac{1}{2L} \sum_{l=1}^{L} p_l(1 - p_l)\mathbb{E}_{\boldsymbol{W}}[g_l(\boldsymbol{W})]$ is maximal for the uniform mode $\boldsymbol{p}^* = (\bar{L}/L, \ldots, \bar{L}/L)$.*

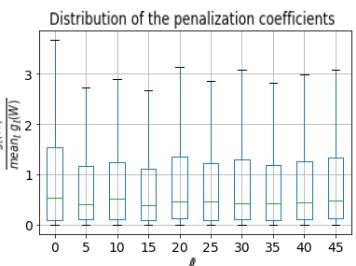

Figure 3: Distribution of $g_l(\boldsymbol{W})$ across the layers at initialization for Vanilla ResNet50 with width 512.

The proof of Theorem 1 is based on some results from the signal propagation theory in deep neural network. We provide an overview of this theory in Appendix A0. Theorem 1 shows that, given a training budget $\bar{L}$ and a randomly initialized ResNet with $\mathcal{N}(0, 2/N)$ and $N$ large, the average (w.r.t $\boldsymbol{W}$) maximal regularization at initialization is almost achieved by the uniform mode. This is because the coefficients $\mathbb{E}_{\boldsymbol{W}}[g_l(\boldsymbol{W})]$ are equal under Assumption 1, which we highlight in Fig. 3. As a result, we would intuitively expect that the uniform mode performs best when the budget $\bar{L}$ is large, e.g. $L$ is large and $\bar{L} \approx L$, since in this case, at each iteration, we update the weights of an overparameterized subnetwork, which would require more regularization compared to the small budget regime. We formalize this intuition in Section 5.

In the next section, we show that $\mathcal{SD}$ is linked to another regularization effect that only occurs in the large depth limit; in this limit, we show that $\mathcal{SD}$ *mimics Gaussian Noise Injection methods by adding Gaussian noise to the pre-activations.*

### 4.2 Stochastic Depth mimics Gaussian noise injection

Recent work by Camuto et al. [2020] studied the regularization effect of Gaussian Noise Injection (GNI) on the loss function and showed that adding isotropic Gaussian noise to the activations $z_l$ improves generalization by acting as a regularizer on the loss. The authors suggested adding a zero mean Gaussian noise parameterized by its variance. At training time $t$, this translates to replacing $z_l^t$ by $z_l^t + \mathcal{N}(0, \sigma_l^2 I)$, where $z_l^t$ is the value of the activations in the $l^{th}$ layer at training time $t$, and $\sigma_l^2$ is a parameter that controls the noise level. Empirically, adding this noise tends to boost the performance by making the model robust to over-fitting. Using similar perturbation analysis as in the previous section, we show that when the depth is large, $\mathcal{SD}$ mimics GNI by implicitly adding *a non-isotropic data-adaptive Gaussian noise to the pre-activations $y_l$ at each training iteration.* We bring to the reader's attention that the following analysis holds throughout the training (it is not limited to the initialization), and does not require the infinite-width regime.

Consider an arbitrary neuron $y_{\alpha L}^i$ in the $(\alpha L)^{th}$ layer for some fixed $\alpha \in (0, 1)$. $y_{\alpha L}^i(x, \boldsymbol{\delta})$ can be approximated using a first order Taylor expansion around $\boldsymbol{\delta} = \mathbf{1}$. We obtain similarly,

$$y_{\alpha L}^i(x, \boldsymbol{\delta}) \approx \bar{y}_{\alpha L}^i(x) + \frac{1}{\sqrt{L}} \sum_{l=1}^{\alpha L} \eta_l \langle z_l, \nabla_{y_l} G_l^i(y_l(x; \mathbf{1})) \rangle \tag{7}$$

where $G_l^i$ is defined by $y_{\alpha L}^i(x; \mathbf{1}) = G_l^i(y_l(x; \mathbf{1}))$, $\eta_l = \delta_l - p_l$, and $\bar{y}_{\alpha L}^i(x) = y_{\alpha L}^i(x, \mathbf{1}) + \frac{1}{\sqrt{L}} \sum_{l=1}^{\alpha L} (p_l - 1) \langle z_l, \nabla_{y_l} G_l^i(y_l(x; \mathbf{1})) \rangle \approx y_{\alpha L}^i(x, \boldsymbol{p})$.

Let $\gamma_{\alpha, L}(x) = \frac{1}{\sqrt{L}} \sum_{l=1}^{\alpha L} \eta_l \langle z_l, \nabla_{y_l} G_l^i(y_l(x; \mathbf{1})) \rangle$. With $\mathcal{SD}$, $y_{\alpha L}^i(x; \boldsymbol{\delta})$ can therefore be seen as a perturbed version of $y_{\alpha L}^i(x; \boldsymbol{p})$ (the pre-activation of the average network) with noise $\gamma_{\alpha, L}(x)$. The scaling factor $1/\sqrt{L}$ ensures that $\gamma_{\alpha, L}$ remains bounded (in $\ell_2$ norm) as $L$ grows. Without this scaling, the variance of $\gamma_{\alpha, L}$ will generally explode. The term $\gamma_{\alpha, L}$ captures the randomness of the binary mask $\boldsymbol{\delta}$, which up to a factor $\alpha$, resembles to the scaled mean in Central Limit Theorem(CLT) and can be written as $\gamma_{\alpha, L}(x) = \sqrt{\alpha} \times \frac{1}{\sqrt{\alpha L}} \sum_{l=2}^{\alpha L} X_{l,L}(x)$ where $X_{l,L}(x) = \eta_l \langle z_l, \nabla_{y_l} G_l^i(y_l(x; \mathbf{1})) \rangle$. Ideally, we would like to apply CLT to conclude on the Gaussianity of $\gamma_{\alpha, L}(x)$ in the large depth limit. However, the random variables $X_l$ are generally not $i.i.d$ (they have different variances) and they also depend on $L$. Thus, standard CLT argument fails. Fortunately, there is a more general form of CLT known as Lindeberg's CLT which we use in the proof of the next theorem.

**Theorem 2.** *Let* $x \in \mathbb{R}^d$, $X_{l,L}(x) = \eta_l \mu_{l,L}(x)$ *where* $\mu_{l,L}(x) = \langle z_l, \nabla_{y_l} G_l^i(y_l(x; \mathbf{1})) \rangle$, *and* $\sigma_{l,L}^2(x) = Var_\delta[X_{l,L}(x)] = p_l(1 - p_l)\mu_{l,L}(x)^2$ *for* $l \in [L]$. *Assume that*

1. *There exists* $a \in (0, 1/2)$ *such that for all* $L$, *and* $l \in [L]$, $p_l \in (a, 1 - a)$.

2. $\lim_{L \to \infty} \frac{\max_{k \in [L]} \mu_{k,L}^2(x)}{\sum_{l=1}^L \mu_{l,L}^2(x)} = 0$.

3. $v_{\alpha, \infty}(x) := \lim_{L \to \infty} \frac{\sum_{l=1}^L \sigma_{l,L}^2(x)}{L}$ *exists and is finite.*

*Then,* $\quad \gamma_{\alpha, L}(x) \xrightarrow[L \to \infty]{D} \mathcal{N}(0, \alpha\, v_{\alpha, \infty}(x))$.

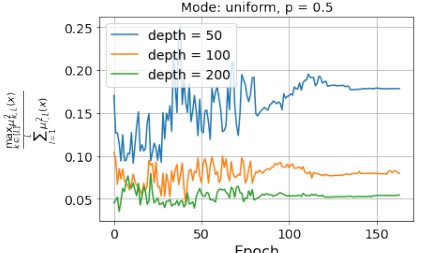

Figure 4: (Theorem 2) Assumption 2 as a function the depth $L$ and epoch.

Fig. 4 provides an empirical verification of the second condition of Theorem 2 across all training epochs. There is a clear downtrend as the depth increases; this trend is consistent throughout training, which supports the validity of the second condition in Theorem 2 at all training times. Theorem 2 shows that training a ResNet with $\mathcal{SD}$ involves implicitly adding the noise $\gamma_{\alpha, L}(x)$ to $y_{\alpha L}^i$. This noise becomes asymptotically normally distributed[3], confirming that $\mathcal{SD}$ implicitly injects input-dependent Gaussian noise in this limit. Camuto et al. [2020] studied GNI in the context of input-independent noise and concluded on the benefit of such methods on the overall performance of the trained network. We empirically confirm the results of Theorem 2 in Section 6 using different statistical normality tests.

Similarly, we study the *implicit* regularization effect of $\mathcal{SD}$ induced on the gradient in Appendix A3, and show that under some assumptions, $\mathcal{SD}$ acts implicitly on the gradient by adding Gaussian noise in the large depth limit.

## 5 The Budget Hypothesis

We have seen in Section 4 that given a budget $\bar{L}$, the uniform mode is linked to maximal regularization with $\mathcal{SD}$ at initialization (Theorem 1). Intuitively, for fixed weights $\boldsymbol{W}$, the magnitude of standard regularization methods such as $\|.\|_1$ or $\|.\|_2$ correlates with the number of parameters; the larger the model, the bigger the penalization term. Hence, in our case, we would require the regularization term to correlate (in magnitude) with the number of parameters, or equivalently, the number of trainable layers. Assuming $L \gg 1$, and given a fixed budget $\bar{L}$, the number of trainable layers at each training

---

[3]The limiting variance $v_{\alpha, \infty}(x)$ depends on the input $x$, suggesting that $\gamma_{\alpha, L}(.)$ might converge in distribution to a Gaussian process in the limit of large depth, under stronger assumptions. We leave this for future work.

iteration is close to $\bar{L}$ (Lemma 1). Hence, the magnitude of the regularization term should depend on how large/small the budget $\bar{L}$ is, as compared to $L$.

**Small budget regime ($\bar{L}/L \ll 1$).** In this regime, the effective depth $L_\delta$ of the subnetwork is small compared to $L$. As the ratio $\bar{L}/L$ gets smaller, the sampled subnetworks become shallower, suggesting that the regularization need not be maximal in this case, and therefore $\boldsymbol{p}$ should not be uniform in accordance with Theorem 1. Another way to look at this is through the bias-variance trade-off principle. Indeed, as $\bar{L}/L \to 0$, the variance of the model decreases (and the bias increases), suggesting less regularization is needed. The increase in bias inevitably causes a deterioration of the performance; we call this the Budget-performance trade-off. To derive a more sensible choice of $\boldsymbol{p}$ for small budget regimes, we introduce a new *Information Discrepancy* based algorithm (Section 4). We call this algorithm *Sensitivity Mode* or briefly *SenseMode*. This algorithm works in two steps:

1. Compute the sensitivity ($\mathcal{S}$) of the loss w.r.t the layer at initialization using the approximation,

$$\mathcal{S}_l = \mathcal{L}(\boldsymbol{W}; \mathbf{1}) - \mathcal{L}(\boldsymbol{W}; \mathbf{1}_l) \approx \nabla_{\delta_l} \mathcal{L}(\boldsymbol{W}; \boldsymbol{\delta})_{|\boldsymbol{\delta}=\mathbf{1}}.$$

   $\mathcal{S}_l$ is a measure of the sensitivity of the loss to keeping/removing the $l^{th}$ layer.

2. Use a mapping $\varphi$ to map $\mathcal{S}$ to the mode, $\boldsymbol{p} = \varphi(\mathcal{S})$, where $\varphi$ is a linear mapping from the range of $S$ to $[p_{min}, 1]$ and $p_{min}$ is the minimum survival rate (fixed by the user). $\phi$ is the linear mapping from the range of $S$ to the segment $[p_{min}, 1]$. In other words, $p_l = p_{min} + \alpha \times S_l$, where the constant alpha is chosen in order to satisfy the budget constraint: $\sum_l p_l = \tilde{L}$.

**Large budget regime ($\bar{L}/L \sim 1$).** In this regime, the effective depth $L_\delta$ of the subnetworks is close to $L$, and thus, we are in the overparameterized regime where maximal regularization could boost the performance of the model by avoiding over-fitting. Thus, we anticipate the uniform mode to perform better than other alternatives in this case. We are now ready to formally state our hypothesis,

**Budget hypothesis.** *Assuming $L \gg 1$, the uniform mode outperforms SenseMode in the large budget regime, while SenseMode outperforms the uniform mode in the small budget regime.*

We empirically validate the Budget hypothesis and the Budget-performance trade-off in Section 6.

## 6 Experiments

The objective of this section is two-fold: we empirically verify the theoretical analysis developed in sections 3 and 4 with a Vanilla ResNet model on a toy regression task; we also empirically validate the Budget Hypothesis on the benchmark datasets CIFAR-10 and CIFAR-100 [Krizhevsky et al., 2009]. Notebooks and code to reproduce all experiments, plots and tables presented are available in the supplementary material. We perform comparisons at constant training budgets.

**Implementation details:** Vanilla Stable ResNet is composed of identical residual blocks each formed of a Linear layer followed by ReLU. ResNet110 follows [He et al., 2016, Huang et al., 2016]; it comprises three groups of residual blocks; each block consists of a sequence Convolution-BatchNorm-ReLU-Convolution-BatchNorm. We use the adjective "Stable" (Stable Vanilla ResNet, Stable ResNet110) to indicate that we scale the blocks using a factor $1/\sqrt{L}$ as described in Section 3. We build on an open-source implementation of standard ResNet[4]. The toy regression task consists of estimating the function $f_\beta : x \mapsto \sin(\beta^T x)$, where the inputs $x$ and parameter $\beta$ are in $\mathbb{R}^{256}$, sampled from a standard Gaussian. CIFAR-10, CIFAR-100 contain 32-by-32 color images, representing respectively 10 and 100 classes of natural scene objects. We present here our main conclusions. Further implementation details and other insightful results are in the Appendix A4.

**Gaussian Noise Injection:** We proceed by empirically verifying the Gaussian behavior of the neurons as described in Theorem 4. For each input $x$, we sample 200 masks and the corresponding $y(x; \boldsymbol{\delta})$. We then compute the p-value $pv_x$ of the Shapiro-Wilk test of normality [Shapiro and Wilk, 1965]. In Fig. 5 we represent the distribution of the p-values $\{pv_x \mid x \in \mathcal{X}\}$. We can see that the Gaussian behavior holds throughout training (left). On the right part of the Figure, we can see that the Normal behavior becomes accurate after approximately 20 layers. In the Appendix we report further experiments with different modes, survival rates, and a different test of normality to verify both Theorem 2 and the critical assumption 2.

---

[4]https://github.com/felixgwu/img_classification_pk_pytorch

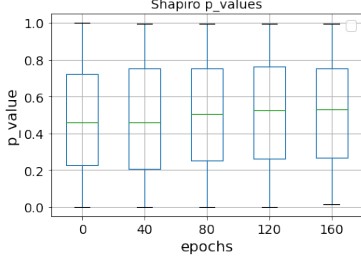
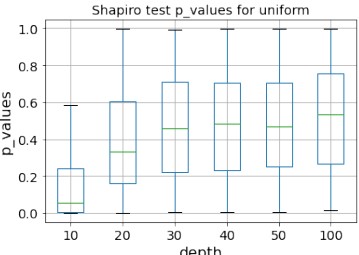

Figure 5: Empirical verification of Theorem 2 on Vanilla ResNet100 with width 128 with average survival probability $\bar{L}/L = 0.7$ and uniform mode. Distribution of the p-values for Shapiro's normality test as a function of the training epoch (left) and depth of the network (right). The tests are performed for the final output neuron $y_L(x)$ (left) and for an arbitrary neuron per layer (right).

**Empirical verification of the Budget Hypothesis:** We compare the three modes: Uniform, Linear, and SenseMode on two benchmark datasets using a grid survival proportions. The values and standard deviations reported are obtained using four runs. For SenseMode, we use the simple rule $p_l \propto |\mathcal{S}_l|$, where $\mathcal{S}_l$ is the sensitivity (see section 5). We report in Table 2 the results for Stable ResNet110. Results with Stable ResNet56 are reported in Appendix A4.

The empirical results are coherent with the Budget Hypothesis. When the training budget is large, i.e. $\bar{L}/L \geq 0.5$, the Uniform mode outperforms the others. We note nevertheless that when $\bar{L}/L \geq 0.9$, the Linear and Uniform models have similar performance. This seems reasonable as the Uniform and Linear probabilities become very close for such survival proportions. When the budget is low, i.e. $\bar{L}/L < 0.5$, the SenseMode outperforms the uniform one (the linear mode cannot be used with budgets $\bar{L} < L/2$ when $L \gg 1$, since $\sum p_l/L > 1/2 - 1/(2L) \sim 1/2$), thus confirming the Budget hypothesis. Table 2 also shows a clear Budget-performance trade-off.

Table 2: Comparison of the modes of selection of the survival probabilities with fixed budget with Stable ResNet110.

| $\bar{L}/L$ | Uniform | SenseMode | Linear |
|---|---|---|---|
| 0.1 | $17.2 \pm 0.3$ | $\mathbf{15.4} \pm 0.4$ | – |
| 0.2 | $10.3 \pm 0.4$ | $\mathbf{9.3} \pm 0.5$ | – |
| 0.3 | $7.7 \pm 0.2$ | $\mathbf{7.0} \pm 0.3$ | – |
| 0.4 | $\mathbf{7.4} \pm 0.3$ | $7.3 \pm 0.4$ | – |
| 0.5 | $\mathbf{6.8} \pm 0.1$ | $7.3 \pm 0.2$ | $9.1 \pm 0.1$ |
| 0.6 | $\mathbf{6.3} \pm 0.2$ | $6.9 \pm 0.1$ | $7.5 \pm 0.2$ |
| 0.7 | $\mathbf{5.9} \pm 0.1$ | $7.3 \pm 0.3$ | $6.4 \pm 0.2$ |
| 0.8 | $\mathbf{5.7} \pm 0.1$ | $6.6 \pm 0.2$ | $6.1 \pm 0.2$ |
| 0.9 | $\mathbf{5.7} \pm 0.1$ | $6.2 \pm 0.2$ | $6.0 \pm 0.2$ |
| 1 | | $6.37 \pm 0.12$ | |

(a) CIFAR10

| $\bar{L}/L$ | Uniform | SenseMode | Linear |
|---|---|---|---|
| 0.1 | $55.2 \pm 0.4$ | $\mathbf{51.3} \pm 0.6$ | – |
| 0.2 | $38.3 \pm 0.3$ | $\mathbf{36.4} \pm 0.4$ | – |
| 0.3 | $31.4 \pm 0.2$ | $\mathbf{30.1} \pm 0.5$ | – |
| 0.4 | $30.9 \pm 0.2$ | $\mathbf{28.5} \pm 0.4$ | – |
| 0.5 | $\mathbf{28.4} \pm 0.3$ | $29.5 \pm 0.5$ | $36.5 \pm 0.4$ |
| 0.6 | $\mathbf{26.5} \pm 0.4$ | $29.9 \pm 0.6$ | $30.9 \pm 0.4$ |
| 0.7 | $\mathbf{25.8} \pm 0.1$ | $29.5 \pm 0.3$ | $27.3 \pm 0.3$ |
| 0.8 | $\mathbf{25.5} \pm 0.1$ | $30.0 \pm 0.3$ | $25.7 \pm 0.2$ |
| 0.9 | $\mathbf{25.5} \pm 0.3$ | $28.3 \pm 0.2$ | $\mathbf{25.5} \pm 0.2$ |
| 1 | | $26.5 \pm 0.2$ | |

(b) CIFAR100

## 7 Related work

The regularization effect of Dropout in the context of linear models has been the topic of a stream of papers [Wager et al., 2013, Mianjy and Arora, 2019, Helmbold and Long, 2015, Cavazza et al., 2017]. This analysis has been recently extended to neural networks by Wei et al. [2020] where authors used a similar approach to ours to depict the explicit and implicit regularization effects of Dropout. To the best of our knowledge, our paper is the first to provide analytical results for the regularization effect of $\mathcal{SD}$, and study the large depth behaviour of $\mathcal{SD}$, showing that the latter mimics Gaussian Noise Injection [Camuto et al., 2020]. A further analysis of the implicit regularization effect of $\mathcal{SD}$ is provided in Appendix A3.

## 8 Limitations and extensions

In this work, we provided an analytical study of the regularization effect of $\mathcal{SD}$ using a second order Taylor approximation of the loss function. Although the remaining higher order terms are usually dominated by the second order approximation (See the quality of the second order approximation in Appendix A4), they might also be responsible for other regularization effects. This is a common limitation in the literature on noise-based regularization in deep neural networks [Camuto et al., 2020, Wei et al., 2020]. Further research is needed to isolate the effect of higher order terms.

We also believe that SenseMode opens an exciting research direction knowing that the low budget regime has not been much explored yet in the literature. We believe that one can probably get even better results with more elaborate maps $\varphi$ such that $\boldsymbol{p} = \varphi(\mathcal{S})$. Another interesting extension of an algorithmic nature is the dynamic use of *SenseMode* throughout training. We are currently investigating this topic which we leave for future work.

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
