# OpenReview forum: "Regularization in ResNet with Stochastic Depth"
_NeurIPS.cc/2021/Conference — NeurIPS 2021 Poster_

### Official Review · Reviewer_j3iT · 2021-07-02

**Rating:** 7
**Confidence:** 4

**Summary:**


The authors study stochastic depth from an theoretical point of view. In particular, they provide insights into the effective depth at each iteration (Lemma 1), discuss the relationship of stochastic depth with exploding gradients (Proposition 1), and relate stochastic depth with explicit regularization (Section 4.1) and Gaussian noise injection (Section 4.2).

**Limitations And Societal Impact:**

Yes.

**Main Review:**

Their results are novel and provide first insights into stochastic depth from a mathematical perspective. One could argue that the results require many assumptions (such as the abundant, not always completely obvious approximations) and only scratch at the surface of stochastic depth (for example, what do these regularizers mean?). But a complete analytical understanding of stochastic depth, in my opinion, is currently out of reach, and the authors provide a first, reasonable step toward it.

I have not read the proofs in the Appendix.

There is some work on layer-wise regularization that might be connected especially to Section 4.1:

Hebiri, M. and Lederer, J., 2020. Layer sparsity in neural networks. arXiv preprint arXiv:2006.15604.



**Time Spent Reviewing:**

2

---

> ### Author Response · Authors · 2021-08-09
> **Response**
>
> We thank the reviewer for the positive feedback and for mentioning this reference that we missed. We have now included it in the list of references.

---

### Official Review · Reviewer_AyjF · 2021-07-16

**Rating:** 6
**Confidence:** 4

**Summary:**

The paper focuses on understanding the regularization effect of stochastic depth in residual neural networks. The authors give 3 main results:

(a) Stochastic depth helps to mitigate (to some extent) the gradient explosion phenomena observed at initialization in the Vanilla ResNet model,

(b) Stochastic depth mimics Gaussian Noise Injection (GNI) in deeper models by implicitly adding some non-isotropic Gaussian noise at each layer during training. Thereby, they show that Stochastic depth has the same regularization effects as GNI.

(c) They propose SenseMode, a principled method to choose the survival rates of each layer for training. SenseMode performs better than the existing methods in certain cases.

The authors corroborate all their propositions in an experimental study.

**Limitations And Societal Impact:**

I have mentioned different issues that exist in the main paper in my main review. The authors discuss the remaining limitations pretty well. As the authors have mentioned, the paper focuses on the general analysis of a regularization method. Hence I don't see a potential negative societal impact.

**Main Review:**



The main paper needs a lot of work before it can be published. Here are my questions and few suggestions.

(1) How do you define the linear mapping $\varphi$ in SenseMode? Is the definition arbitrary? Or do you sort the values and match them to equally spaced numbers between $[p_{\min}, 1]$? How does this kind of mapping solve the issue in the small budget regime when the regularization contains the term $p_{l} (1 - p_{l})$ (which is a quadratic function)?

Also, it will be great to have similar experiments for Vanilla ResNet in table 2. Because all the propositions and experiments involved Vanilla ResNet and SenseMode was built on the structure of the Vanilla ResNet model.

(2) How does proposition 1 change when you have batch norm in the network? Will you still observe exponential growth of $\tilde{q}_{\ell}(x, z)$? Or is it just going to be a linear growth? Is the $1/\sqrt{L}$ factor still necessary with the batch norm? Because I see in [1], the authors also consider Vanilla ResNet while proposing the $1/\sqrt{L}$ factor? It will be interesting to see if explosion still exists with batch-norm in experiments and stochastic depth still reduces exponential growth. Otherwise, I don't see a point in Proposition 1.

(3) How do I read the plot in Figure 5? Can you please give more details in the experimental section about "Gaussian Noise Injection"? For example, what neurons do you consider for the left plot in Figure 5? Do you take the entire set of neurons to do the left plot and all the neurons per layer for the right plot?

(4) Multiple issues in notation and presentation:

(a) What is $G_{l}$? The authors need to specify the function clearly. Is it the function that connects $y_l$ to $y_{out}$? It's not the way to define a function, as has been mentioned in the paper.

(b) Use of both $l$ and $\ell$ is frustrating. For example, the reader has to differentiate the $l$ and $\ell$ used in $H_{\ell}$ and $\zeta_{l}$ (equation 5).

(c) Clearly demarcate Vanilla ResNet and Stable ResNet110. It will be good to have some diagram differentiating the two. As far as I have understood (and reviewed), Vanilla ResNet contains only linear layers and ReLU in their residual blocks. In contrast, Stable ResNet110 contains the sequence BatchNorm-ReLU -Convolution-BatchNorm in their residual blocks.  It's essential to clearly demarcate this since experiments in Figure 2, Figure 3, Figure 4, Figure 5 are done with Vanilla ResNet, and the experiments in Table 2 are done with Stable Resnet110. Also, at some places, you have mentioned Vanilla Stable Resnet. Would you please demarcate the terms properly?

(d) The caption in table 1 says that the table reports $\tilde{q}_l(x, z)$ at initialization. However, I don't see exponential numbers as claimed by proposition 1. Did you mean to write the growth rate (as mentioned in the text)?

(5) Plots lack proper labels and captions:

(a) Figure 1(b) doesn't have a label. What do the two lines in the graph refer to?

(b) What are the numbers in Table 2(a) and (b)? Are they error or accuracy of the models in the task?

(c) The x and y labels in all the plots are unreadable. I had to guess every time I read some plot (from a printed pdf).




1: S. Hayou, E. Clerico, B. He, G. Deligiannidis, A. Doucet, and J. Rousseau. Stable resnet. In Proceedings of The 24th International Conference on Artificial Intelligence and Statistics, pages 324–1332, 2021a.



***After Rebuttal***
I have increased my score to 6. The paper has many clarity issues and I would appreciate if the authors could incorporate changes to address the concerns raised by all the reviewers.

**Time Spent Reviewing:**

3 - 5 hours

---

> ### Author Response · Authors · 2021-08-09
> **Response**
>
> We thank the reviewer for their feedback. We have addressed all their suggestions and questions, which were easy to incorporate in the revised version of the paper.
>
> 1) "How do you define the linear mapping φ in SenseMode? Is the definition arbitrary? Or do you sort the values and match them to equally spaced numbers between [pmin,1]
> ? How does this kind of mapping solve the issue in the small budget regime when the regularization contains the term pl(1−pl)  (which is a quadratic function)?"
>
> As stated in line 285, \phi is the linear mapping from the range of S to the segment [p_min, 1]. In other words, p_l = p_min + alpha*S_l, where the constant alpha is chosen in order to satisfy the budget constraint: \sum_l p_l = \tilde L.
> In the small budget regime (p_l close to zero), the regularization term p_l(1-p_l) is dominated by p_l. Therefore, in this regime, SenseMode is a simple procedure that allows to increase at the same time the expressivity of the model (captured layerwise by S_l) and the regularization term.
>
> - "Also, it will be great to have similar experiments for Vanilla ResNet in table 2. Because all the propositions and experiments involved Vanilla ResNet and SenseMode was built on the structure of the Vanilla ResNet model."
>
> This is a good suggestion. We ran the same experiments in table 2 on the datasets MNIST and Fashion-MNIST as they are more suited for the Vanilla Model (Cifar10 and 100 are too complex for this toy model). On MNIST, the different modes give the same performance, going from 96% accuracy with a budget \tilde L/ L = 0.1, to 98% when the budget \tilde L/ L >= 0.7. We believe that this dataset might be too simple to distinguish between the modes. On Fashion-MNIST, we see a similar behaviour as described in the main text (the error rates are reported in the table below). We added these results to the appendix.
>
> +-----+-----------+---------+
>
> |      | SenseMode | Uniform |.
>
> +-----+-----------+---------+
>
> | 0.1 |    19.8   |   22.1  |
>
> +-----+-----------+---------+
>
> | 0.2 |    12.0   |   12.9  |
>
> +-----+-----------+---------+
>
> | 0.3 |    11.5   |    12   |
>
> +-----+-----------+---------+
>
> | 0.4 |    11.5   |   11.5  |
>
> +-----+-----------+---------+
>
> | 0.5 |    11.8   |   11.8  |
>
> +-----+-----------+---------+
>
> | 0.6 |    12.7   |   11.7  |
>
> +-----+-----------+---------+
>
> | 0.7 |    11.9   |   11.4  |
>
> +-----+-----------+---------+
>
> | 0.8 |    11.5   |   11.5  |
>
> +-----+-----------+---------+
>
> | 0.9 |    11.4   |   11.2  |
>
> +-----+-----------+---------+
>
> | 1.0 | 12.0      | 12.0   |
>
> +-----+---------------------+
>
> 2) "How does proposition 1 change when you have batch norm in the network? Will you still observe exponential growth of q_ℓ(x,z)? Or is it just going to be a linear growth? Is the 1/sqt{L} factor still necessary with the batch norm? Because I see in [1], the authors also consider Vanilla ResNet while proposing the 1/sqrt{L} factor? It will be interesting to see if explosion still exists with batch-norm in experiments and stochastic depth still reduces exponential growth. Otherwise, I don't see a point in Proposition 1."
>
> Edit: The result of proposition 1 holds without BatchNorm. During training, BatchNorm helps avoid the gradient exploding (prop 1 does not hold during training). All our experiments in table 2 are done with BatchNorm, which confirms our theoretical insights even with BatchNorm.
>
> 3) "How do I read the plot in Figure 5? Can you please give more details in the experimental section about "Gaussian Noise Injection"? For example, what neurons do you consider for the left plot in Figure 5? Do you take the entire set of neurons to do the left plot and all the neurons per layer for the right plot?"
>
> For the left plot of Figure 5, the test is performed for the final output neuron y_L(x). For the right plot, we consider an arbitrary neuron per layer. We added a comment in the main text.
>
> 4) "Multiple issues in notation and presentation:"
>
> (a) "What is Gl?"
>
> G_l is defined “implicitly” as the function that maps y_l to y_out. We have updated the paper to make this clear.
>
> (b) "Use of both l and ℓ is frustrating. For example, the reader has to differentiate the l and ℓ used in Hℓ and ζl (equation 5)."
>
>  ℓ stands for the loss function and l for the layer. H_ℓ is therefore the Hessian of the loss ℓ. We changed the notation to H to avoid any confusion.
>
> (c) "Clearly demarcate Vanilla ResNet and Stable ResNet110...Would you please demarcate the terms properly?"
>
> We agree with the reviewer that it is crucial to demarcate Vanilla ResNet and ResNet110, which is what we try to do in Section 2 (line 47), in Section 6 (lines 300-304), and in the supplementary material (Appendix A4).  Following the reviewer's recommendation, we reformulated the explanation to make it more clear:
> Vanilla ResNet is composed of identical residual blocks, each formed of a Linear and a ReLu layer. ResNet110 follows [He et al., 2016, Huang et al., 2016]; it comprises three groups of residual blocks; each block consists of a sequence of layers Convolution-BatchNorm-ReLU-Convolution-BatchNorm. We use the adjective 'Stable' (Stable Vanilla ResNet, Stable ResNet110) to indicate that we scale the blocks using a factor 1/sqrt(L) as described in Section 3.
>
> This scaling has been shown to avoid gradient explosion at initialization (see [1]), justifying the choice of the adjective “Stable”.
>
> (d) "The caption in table 1 says that the table reports q~l(x,z) at initialization. However, I don't see exponential numbers as claimed by proposition 1. Did you mean to write the growth rate (as mentioned in the text)?"
>
> We thank the reviewer for spotting this typo, we indeed meant to write the growth rate as mentioned in the text.
>
> (5) "Plots lack proper labels and captions:"
> (a) "Figure 1(b) doesn't have a label. What do the two lines in the graph refer to?"
>
> We have added labels to Fig 5-b (which we forgot but is the same as Figure 5-a).
>
> (b) "What are the numbers in Table 2(a) and (b)? Are they error or accuracy of the models in the task?"
>
> Error rates.
>
> (c) "The x and y labels in all the plots are unreadable. I had to guess every time I read some plot (from a printed pdf)."
>
> We followed the reviewer’s suggestion and increased the font size of the labels.

---

> > ### Comment · Reviewer_AyjF · 2021-08-21
> > **Batch norm**
> >
> > I want to thank the authors for their detailed response. I want to focus on the question about the 'batch norm' that I had asked before.
> >
> > Batch norm normalizes the batch of inputs with variance across the batch. Hence, whatever explosion the network has suffered during forward propagation till block $\ell$ gets normalized with the batch norm in the block $\ell$. I strongly believe the same phenomenon happens with gradients as well. Hence, I believe the explosion will be linear than exponential. I would like to see what the authors think about it.

---

> > > ### Author Response · Authors · 2021-08-21
> > > **Response 2**
> > >
> > > We thank the reviewer for their feedback.
> > >
> > > We would like to emphasize that the result of Prop1 is true only at initialization and without BN. A full characterization of the gradient second moment with BN at initialization is out of the scope of this paper since BN involves computing means and standard deviations which both depend on the initialization weights. Hence, taking the expectation w.r.t the initialization weights does not yield closed form analytic formulas, and different machinery might be needed to study the dynamics of the gradient second moment in this case.
> > >
> > > Our intuition (in the previous answer) follows findings from [2] where the authors show that for feedforward neural networks, BN leads to exploding gradient second moment.
> > > We would also like to emphasize that Prop1 is a secondary result in our paper, and the different regularization effects of SD are the main results.
> > >
> > >
> > > [2] A Mean Field Theory of Batch Normalization. Yang et al. (2019)

---

### Official Review · Reviewer_Z7Sb · 2021-07-16

**Rating:** 6
**Confidence:** 4

**Summary:**

The paper provides theoretical analysis for stochastic depth from a regularization perspective. The paper further illustrates the different regularization effects of stochastic depth based on perturbation analysis and signal propagation. The analysis leads to principled guidelines for choosing the survival rates.


**Limitations And Societal Impact:**

Limitations that are not addressed and suggestions are included in the above main review. The paper really needs proofreading and grammar check.

**Main Review:**

The paper is well written and easy to follow.

The paper discusses interesting regularization effects of stochastic depth such as flatness and information discrepancy.

There is no detail about \phi in Line 284. It is important to provide detailed discussion about how to choose the linear mapping \phi and how it impacts the performance.

Line 314: it is not clear why normal behavior is more accurate for deeper layers under uniform mode with large budget. Does this implies SD only mimics Gaussian Noise Injection for deeper layers?

Section 4 title seems incomplete.

Theorem 4 is not provided in the paper.


**Time Spent Reviewing:**

10

---

> ### Author Response · Authors · 2021-08-09
> **Response**
>
> We are extremely surprised by the low score in this review given the concerns mentioned by the reviewer. We address these concerns below.
>
> 1) "There is no detail about \phi in Line 284. It is important to provide detailed discussion about how to choose the linear mapping \phi and how it impacts the performance."
>
> As stated in line 285, \phi is the linear mapping from the range of S to the segment [p_min, 1]. In other words, p_l = p_min + alpha*S_l, where the constant alpha is chosen in order to satisfy the budget constraint: \sum_l p_l = \tilde L. We have updated the paper to make this more clear.
>
> 2) "Line 314: it is not clear why normal behavior is more accurate for deeper layers under uniform mode with large budget. Does this implies SD only mimics Gaussian Noise Injection for deeper layers?"
>
> This is a direct result of Theorem 2. The Gaussian behaviour is valid in the infinite depth limit, therefore, the more layers you average over, the more accurate is the Gaussian approximation.
>
> 3) "Section 4 title seems incomplete."
>
> Thank you for noticing this typo. We have fixed this in the revised version of the paper.
>
> 4) "Theorem 4 is not provided in the paper."
>
> This should be Thm2 and not Thm4. We have fixed this issue in the updated version of the paper.

---

> > ### Comment · Reviewer_Z7Sb · 2021-08-26
> > **Thank you for the detailed response**
> >
> > I appreciate that my concerns have been addressed. As in my original review, the theoretical contribution to better understand SD and even more general regularization techniques is quite interesting. I would still suggest the authors to improve the clarity of the paper. Proofread it and take care of the notations and definitions to make it readable to wider audience. All in all, I'm happy to raise my score to 6.

---

> ### Author Response · Authors · 2021-08-23
> **Response**
>
> Could you please let us know if our response has adressed your concerns?

---

### Official Review · Reviewer_eU7J · 2021-07-21

**Rating:** 6
**Confidence:** 3

**Summary:**

This paper studies the regularization effects involved in the stochastic depth training of ResNets. Specifically, by applying the second-order Taylor approximation to the loss function and the intermediate features, the authors show that the stochastic depth training involves a regularization term that promotes flatness and acts like Gaussian noise to the features. Based on this theoretical insight, the authors suggest a strategy for setting the survival probabilities of sub-networks depending on the computational budget, which corresponds to controlling the regularization effect. This strategy developed upon the new insight in the paper is validated by empirical experiments.


**Limitations And Societal Impact:**

The authors addressed the limitation in the paper, and I agree with the authors' claim that this work will have no direct societal impact.

**Main Review:**

Stochastic depth is a simple yet effective regularization technique for training very deep ResNets. However, it has been unclear how the stochastic depth induces regularization effects and how to set dropout rates of sub-networks. In this regard, the authors present valuable theoretical analysis that shows a flatness promoting regularization term and layer-wise Gaussian noise effects involved in the stochastic depth training. I think that this novel insight can help to develop effective strategies for stochastic depth training and its novel variants, and as an example, the authors give a strategy for setting dropout rates depending on the computational budget. Considering the prevalence of ResNet in the machine learning community, I believe that this work will be of interest to both researchers and practitioners.

The paper is well-written, and the structure is easy-to-follow and interesting. However, there are few minor typos here and there (such as missing legend in Fig 1 (b), crossentropy (L159), The empirical loss given by (L160), hessian (footnote in p.5)).


* Minor commnet: The authors use the Taylor approximation around the drop ratio of 1, which is highly unlikely in practice and can make higher-order terms non-negligible. I’m wondering if the authors can explain how the small drop rates can impact the regularization term in Eq 6 and Gaussian noise $\gamma_{\alpha, L}(x)$.


**Time Spent Reviewing:**

20

---

> ### Author Response · Authors · 2021-08-09
> **Response**
>
> We thank the reviewer for the positive feedback and comments; we hope that our contribution will indeed interest both researchers and practitioners.
> We also thank the reviewer for spotting the typos and we have corrected them.
> We agree that in practice, the first-order approximation around 1 can sometimes be inaccurate. One can consider higher-order expansions for more accuracy at the cost of simplicity, but this is out of the scope of the current work. We note nevertheless that the empirical experiments tend to validate the first-order approximation results, even for small drop rates. We hope this answers the reviewer’s question.

---

### Decision · Program_Chairs · 2021-09-27

**Decision:**

Accept (Poster)

**Comment:**

This paper gives a theoretical analysis of the stochastic depth (SD) technique for ResNet. It is shown that SD mitigates the gradient explosion phenomenon. Moreover, it is shown that SD behaves like a Gaussian noise injection to the features in the internal layers and thus it works as a regularization. Finally, the authors proposed a method called SenseMode that determines the mode under a small computational budget limitation.

The analysis given in this paper is interesting. It shed light to understand what SD is essentially doing. The proposed SenseMode is also useful.
On the other hand, as the reviewers pointed out, there are some issues in its writing, i.e., typos, confusing notations and unclear descriptions. In particular, the definition of the linear mapping $\varphi$ in the definition of SenseMode should be explicitly given in the final version. I guess everybody would be confused at this point.

In summary, this paper has sufficient novelty and gives a good insight for SD, which is beneficial to the community. I think this paper can be accepted by NeurIPS.